# Reconstructing Attitudes towards Work from Home during COVID-19: A Survey of South Korean Managers

**DOI:** 10.3390/bs11120163

**Published:** 2021-11-27

**Authors:** Patrick Allen Rose, Suzana Brown

**Affiliations:** 1Seoul Business School, Seoul School of Integrated Sciences and Technologies, 46 Ewhayeodae 2-gil, Fintower, Sinchon-ro, Seodaemun-gu, Seoul 03767, Korea; 2Department of Technology and Society, The State University of New York, Korea, 119-2 Songdo Munhwa-ro, Yeonsu-gu, Incheon 21985, Korea; suzana.brown@sunykorea.ac.kr

**Keywords:** work from home, telework, South Korea, COVID-19, policy, job feasibility, attitudes, managers, outcomes, trade-offs, job satisfaction, productivity, trust, culture

## Abstract

This article explores how after almost two years of government-imposed work from home (WFH) for the purpose of curbing the spread of COVID-19, South Korean managers’ general attitudes towards WFH may have been reconstructed and if this change influenced their expectations that WFH would persist for the long run. Before COVID-19, WFH was rare, and the country was well known for having one of the most hierarchical and rigid work cultures, with long hours at the office being the norm. The results of this study are based on survey responses from 229 South Korean managers and executives. Using means comparisons and hierarchical linear multiple regression models to answer three research questions, the present study evaluates theorized predictors of WFH take-up, general attitudes towards WFH, and the likelihood that WFH will continue post-COVID-19. The results indicate that forced WFH adoption during COVID-19 had statistically significant positive effects on the attitudes of South Korean managers and their intentions to continue working from home in the future. This study has practical implications for companies and governments that are interested in taking advantage of WFH and implementing it more permanently. It provides interesting findings on how managers from a country with minimal WFH prior to COVID-19 perceive the benefits of WFH and how they respond to its mandated adoption.

## 1. Introduction and Background

### 1.1. Purpose

The present article investigates if the experience with work from home (WFH) in South Korea may have reconstructed past antagonistic attitudes towards it or alternatively its rushed implementation aggravated negative perceptions rather than reduced them. As a result, it assesses whether managers believe the sudden shift to WFH is here to stay or if they expect to go back to the orthodox office work routine after social distancing guidelines are rescinded. Succinctly, it asks how the so-called experiment with WFH is going from the perspective of South Korean managers. Do they believe the positives outweigh the negatives and does this have an effect on their expectations? In exploring this question, insights are gained into the factors contributing to WFH outcomes.

### 1.2. Background and Context

With the onset of the COVID-19 pandemic in South Korea, the government mandated that organizations reduce exposure to infections among the working population by implementing work-from-home (WFH) policies and allowing workers to have flexible schedules. The mass adoption of WFH technology and practices represents an unprecedented social experiment in South Korea and globally that is poised to change how work is done in the long term. This article is interested in exploring the consequences of this far-reaching so-called ‘social experiment’ during which companies have scrambled to send employees home to work often without much in the way of technical support, skills training, or prior experience with remote work [1].

Before COVID-19, employees who WFH were rare in South Korea. South Korea is known for having one of the most hierarchical and rigid work cultures, in which long hours at the office are customarily seen as an indispensable exhibition of organizational commitment [2,3]. In South Korea, for example, it is often frowned upon for employees to leave the office before their bosses have left and that often means late in the evening. Before COVID-19, South Korea was ranked 3rd among OECD countries for working long hours (average 48 h per week) and only approximately 8% of South Korean managers used flexible working arrangements including WFH [4].

In such a hard-working environment, the last two years have been a major shift as most organizations in South Korea were forced to enact WFH policies. Statista reports that up to 67% of South Korean employees in certain industries are working from home during the COVID-19 pandemic [5]. This trend is consistent with several other recent reports that have found WFH is substantially increasing globally as a result of companies complying with social distancing mandates during the COVID-19 pandemic [6,7,8]. An international survey of executives found that 58% of all employees work eight or more days each month from home [9]. A recent survey of American employees reported that 85% worked from home after the pandemic started, compared with approximately 10% prior to it [10].

The research literature on the impact of COVID-19 on WFH outcomes is already extensive. A comprehensive literature review by [11] found fifty-five published articles on the impact of WFH by researchers from various disciplines including psychology, social science, economics, finance, and business. The themes covered in the literature are wide-ranging; however, there are only a handful of studies that examine how the recent extensive shift to WFH has restructured managers’ sentiments. Furthermore, none of the studies were conducted with South Korean managers or workers.

### 1.3. Approach

The aim of this article is to bridge the above-described gap and investigate how the unprecedented mass adoption of WFH during COVID-19 has disrupted traditional values among South Korean managers about the importance of going to the office. It focuses on whether the lengthy and continued forced experience with WFH during COVID-19 has changed beliefs, attitudes, and practices for the long term. It is believed that attitudes arise from- and are reinforced by experience. The term ‘attitude’ is defined as a learned tendency to evaluate a particular thing as favorable or unfavorable that causes an individual to behave towards the thing in a particular way [12,13,14]. 

The well-known Hierarchy of Effects (HOE) theory suggests that individual decision making is the result of a progression of learning built on experiences that include: cognitive (thinking), affective (feeling), and conative (intent) developmental steps [15]. Individuals are first exposed to and gain awareness about a new practice or technology, subsequently, develop positive or negative attitudes about it, and then make the choice to use it or avoid it [16]. The current article theorizes that positive experiences with WFH (even if imposed) will increase positive feelings toward it if the experience is perceived to be beneficial. Conversely, it follows that negative experiences with WFH will have the opposite effect. It is assumed that managers who have adjusted to WFH and learned to enjoy it will favor WFH and want to continue it even after COVID-19 subsides, but those who view the drawn-out shift to WFH as harmful will oppose it and want to return to the old, normal for Korea, extended workday at the office. 

The research prior to COVID-19 postulated that employees who perceive more advantages and benefits to WFH are more likely to favor it, and conversely, those who report more problems are more likely to be dissatisfied with it [17,18,19,20,21]. Similarly, for organizations and managers, WFH represents both opportunities and risks that need to be weighed and used accordingly [22,23,24,25,26].

### 1.4. Constraints

Despite the advantages and disadvantages of WFH, not every employee is able to WFH even if they want to. Most jobs in certain non-high-tech and information-centered industries cannot be carried out with digital technologies. To understand the causes underlying WFH adoption rates, it is important to assess why employees do not report working from home [7]. The reason could be because of their choice, but it might also be that WFH is largely not feasible because employees’ jobs simply cannot be done at home or organizational policies do not permit it. It is generally accepted, for example, that Knowledge workers as a group are simply more able to WFH compared to other employees in service industries where their jobs require face-to-face contact with customers [27].

The encouragement employees receive from their organizations to work remotely is often believed to be at the root of employees’ ability and willingness to WFH. Lim and Teo [28] explain: “...individuals who perceive that their [managers] are supportive of teleworking will have a more favorable attitude towards teleworking” and they found in their study on Chinese workers that “...their supervisors” understanding, concern, and acceptance of them teleworking was valuable and important to their teleworking attitudes.” Other studies also emphasize the important role of managers’ support for WFH in promoting positive attitudes towards WFH [29,30,31]. 

Given the hierarchical corporate culture that is at the center of South Korean working life, the present article theorizes that management culture is a key construct that mediates WFH take-up. In the current study, the term ‘management culture’ refers to measures of traditional and autocratic leadership styles. It is supposed that managers’ stalwart attitudes, authoritarian practices, workaholism traits, and technostress inhibit WFH participation in organizations [32,33,34]. The impact of organizational support and commitment on creating a work environment that is conducive to WFH is demonstrated by many previous studies [35,36,37].

## 2. Methods

### 2.1. Objectives

The present study explores whether or not the COVID-19 situation has provided good examples to South Korean company managers that flexible remote work from home is a beneficial new way of doing business and thus has reconstructed their attitudes. This research is also interested in investigating how managers’ overall support or opposition to WFH may influence their decisions to continue WFH. Moreover, it considers whether the sudden switch to WFH for almost all employees will result in a permanent change or is it expected that they will be required to return to the office full time after COVID-19 subsides.

### 2.2. Research Questions

Formally, the three primary research questions are:RQ1Has WFH become more prevalent in South Korea during the government’s social distancing mandate to reduce the number of workers in the office, and does organizational WFH policies, job feasibility, and management culture influence this outcome?RQ2Have managers changed their general attitudes towards WFH due to their forced experience with it and does their job satisfaction mediate their response?RQ3Do managers expect WFH to continue at their organizations after the COVID-19 pandemic subsides and what is the relationship between their general attitudes towards WFH and their expectations for WFH?

### 2.3. Participants and Procedure

In August–September 2021, an anonymous self-administered online survey was conducted with managers and executives from South Korean companies about their experience with WFH during the COVID-19 pandemic. The survey included 32 questions and took participants an average of thirteen (13) minutes to complete. The sample frame consisted of contacts with business school alumni from five different South Korean universities, and the sample design used the commonly employed snowball method where respondents provided referrals among their colleagues to invite additional participants. This method was chosen due to restricted information privacy laws in South Korea that prohibit universities from sharing the contact information of alumni.

A completely anonymous survey link was provided for the self-administered online survey. The invitation email and the consent script specified that this study was targeting managers. The survey produced 229 complete responses. The Institutional Review Board of the State University of New York-Korea (SUNY-Korea) granted approval for this study (Approval Number: 202103-IRB-01). The privacy of research participants was protected, and the results are presented only in an aggregated form.

To verify the representativeness of respondents, we used data collected on demographic and background characteristics. There were approximately 20% more male than female participants in the sample (134 vs. 91, respectively). This difference reflects the imbalanced ratio of male to female managers generally found in South Korean organizations. A sizable majority of respondents (89%) had earned a college degree, which would be expected among managers. Likewise, most respondents were over age 30 (86%) and the largest group was over 50 years old (43%). A segment of survey participants only supervised themselves (36%). However, many in this group worked for small organizations (29%) and it likely that the others were self-employed, were not currently working or belonged to organizations with ‘flat’ structures and did not have any direct reports. Regardless, mean comparisons showed that these two groups were not statistically significantly different.

### 2.4. Instrument and Measures

Specific items included in the survey instrument were validated by previous studies identified during an extensive review of the literature on factors influencing work-from-home (WFH) adoption and outcomes, as summarized in the previous introduction to this article. The survey instrument consisted of questions on the rates of WFH take-up in South Korea, reasons why employees can not WFH (job feasibility), organizational policies about WFH, managerial culture, job satisfaction, WFH positive and negative trade-offs, and general attitudes about WFH stemming from recent experiences. Finally, the survey assessed the likelihood that WFH would continue post-COVID-19.

**WFH Take-up**. To verify that South Korean employees were actually switching to WFH after the onset of the COVID-19 crisis, the survey included two items adapted from the PwC’s US Remote Work Survey [38]: The items capture two dimensions of the WFH adoption construct: first, how often managers themselves (individual-level) WFH on average; and second how often did all employees (organizational-level) at the company on average WFH, measured in days per week (none, 1–3 days per month, 1, 2, 3 or 4 days per week, and 100% full time). To measure if WFH has increased since COVID-19, respondents were asked to recall WFH take-up levels prior to the COVID-19 pandemic for themselves personally and their organizations in general. Furthermore, respondents reported if they expected WFH to continue after the pandemic is contained (e.g., “On average, about how much did you personally WFH before COVID-19, do you currently WFH during the pandemic, and would you like to WFH after it subsides?”). In addition, respondents rated the likelihood that WFH will continue at their organizations after COVID-19 (e.g., “Post-COVID-19, how likely is your organization to offer WFH compared to pre-COVID-19?”).

**Job Feasibility.** To examine the relationship between WFH adoption and the ability to WFH, the survey captured the primary reasons why some managers did not WFH. For this purpose, two items were adopted from the VROOM Digital Survey on Remote Work Post COVID-19 [39]. First, company policies regarding WFH (e.g., “Which of the following statements best matches your organization’s WFH policy?”); and second, employees’ ability to work remotely due to the characteristics of their jobs (e.g., “Generally, how difficult is it for employees at your organization to WFH?”).

**Management Culture.** A subset of Cheng et al. [40] paternalistic leadership-style scale index was adapted to represent the construct management culture to the extent that it was traditional or authoritarian (also verified by [41,42]). The first half of this index measured the prevalence of typical authoritarian behaviors in leaders (e.g., “How often do each of the following statements about the leadership style of managers at your organization generally apply?”) and included four items related to leaders restricting the autonomy of employees (e.g., “Managers ask employees to obey their instructions completely.”) evaluated on a five-point scale (1 = nearly always to 5 = almost never). Drawing on Farh et al. [28], the second half of this index focused on subordinates’ dependence on- and fear of autocratic leaders (e.g., “Reflecting on your work experience, how true is each of the following statements about how employees generally view the leadership style of managers at your organization?”) by including five items (e.g., “Employees have to rely directly on their managers for assistance to complete work.”) evaluated on a five-point scale (1 = almost always true to 5 = almost never true). The internal reliability coefficient (Cronbach’s alpha) for the entire set of management culture items was 0.879.

**Job Satisfaction.** A set of six items focused on the construct of job satisfaction adopted from [43] measured managers’ motivation, job stress, and affective commitment (e.g., “Indicate below your overall satisfaction with different elements of your work experience.”) on a four-point scale (1 = “very satisfied” to 4 = “very dissatisfied”). The internal reliability coefficient (Cronbach’s alpha) for the job satisfaction items was 0.850.

**General Attitude towards WFH.** Three related items adapted from PwC’s US Remote Work Survey (2021) and iHASCO and HSM COVID-19 Survey (2021) were used to measure different dimensions of the construct of a general attitude towards WFH. The prevalence of individual-level support for WFH (the major dependent variable for this study) was evaluated using a four-point Likert scale (1 = strongly in favor WHF to 4 = strongly against WFH). The overall organizational acceptance of WFH was measured (e.g., “In general, have your colleagues reacted positively or negatively to the WFH experience?”) on a four-point scale (1 = very positive to 4 = very negative); and beliefs about organizational attitudes were further explored by measuring the assessed organization-wide success of WFH (e.g., “How successful would you say the shift to WFH because of COVID-19 has been for your organization?”) on a five-point scale (1 = very successful to 5 = very unsuccessful). The internal reliability coefficient (Cronbach’s alpha) for the general attitude items was 0.733.

**Negative and Positive Trade-Offs.** Portions of previously developed index scales [29,44] were adapted to represent the two dimensions (individual and organizational) of the positive and negative trade-offs index, which is theorized to be another measure of the general construct defined above. The first item focused on the top advantages and disadvantages of WFH posed to managers in light of their recent forced WHF experience during COVID-19 (e.g., “Based on your own recent work-related experiences, rate how much each of the consequences listed in the table below either increases or decreases when employees WFH.”). Similarly, the second item measured the trade-offs that commonly need to be addressed by organizations (e.g., “Based on your own recent work-related experiences, rate the direction of each pair of the trade-offs listed in the table below when employees WFH.”). For the first item’s scale index, six common individual trade-offs were assessed in opposite pairs (e.g., concentration vs. distraction) as suggested by [44]. The second index scale was composed of eight organizational trade-offs (e.g., support from colleagues) evaluated on a five-point scale (1 = greatly increases to 5 = greatly decreases). The internal reliability coefficients (Cronbach’s alpha) for the Individual Trade-Offs and Organizational Trade-Offs dimensions were 0.868 and 0.694, respectively. Cronbach’s alpha for the entire set of trade-off items was 0.858.

**Productivity:** A separate trade-off item focused on the construct of productivity. This item was adopted from the PwC’s US Remote Work Survey [38] and it measured perceptions of average employee WFH productivity compared to how it was before (e.g., “In your view, how has the average WFH employee productivity changed compared to when they used to work in the office.”) on a five-point scale (1 = much less productive to 5 = much more productive).

**Trust in Employees to WFH.** Related to the concept of productivity above, one section included four items focusing on assessing managers’ trust in their employees to work independently at home. These items were developed based on a qualitative study by Lombardo and Mierzwan [39] and focus on managers’ perceptions of the effectiveness of remote supervision (e.g., “Employees that WFH are much more difficult to monitor and control.”). They were evaluated on a four-point scale (1 = strongly agree to 4 = strongly disagree). The internal reliability coefficient (Cronbach’s alpha) for this index was 0.741. 

**Control Variables.** Standard demographic characteristics on respondents were collected such as gender, age, marital status, educational attainment, and annual household income in addition to other background identifiers including occupation industry, company size, length of employment, number of people supervised. This information was used to cross-validate this study’s sample by comparing the characteristics of survey respondents to the target sample to confirm it was representative and as controls in hierarchical linear multiple regression models present later in this article.

### 2.5. Latent Constructs

In the present study, management culture is theorized to mediate the relationship between managers’ evaluation of WFH positive and negative trade-offs (perceived outcomes) and their overall attitudes towards WFH. Furthermore, this study explores how overall attitudes towards WFH (specifically measured as to how respondents assess the success of WFH for their organizations) may have been influenced by positive or negative experiences with forced WFH during COVID-19. Figure 1 below presents the conceptual model for data analysis.

### 2.6. Data Analysis

Data were entered into Statistical Package for Social Science Version 26 (SPSS, IBM Corp: Armonk, NY, USA). Confirmatory factor analysis (CFA) was used to evaluate the construct validity and reliability of the index scales and which factors should be retained in each group. In order to check if items in the indices were related to their respective construct and each other, calculations of the collinearity (Cronbach’s alpha) were used to assess the convergent and discriminant consistency of the factors, which are reported earlier in this section for each index. All the indices showed acceptable goodness of fit, as was expected due to their repeated validation by prior research discussed previously.

After the index measures were verified with factor analysis, descriptive statistics summarized data on the reported WFM take-up and general attitudes towards WFH comparing changes before and since the COVID-19 pandemic by demographic subgroups. Means comparisons explored the differences between WFH groups in relation to: (1) organizational WFH policies, (2) respondents’ ability to WFH (i.e. job feasibility), (3) general attitudes towards WFH, (4) job satisfaction, and (5) management culture. Associations were tested between WFH take-up groups and the aforementioned independent variables in relation to general attitudes towards WFH and expectations of the continuation of WFH post-COVID-19.

Three hierarchical linear multiple regression models were developed to evaluate the strengths of the relationships between the independent variables (organizational WFH policy, job feasibility, management culture and job satisfaction) and the three major dependent outcome variables: (1) WFH take-up during COVID-19; (2) general attitude towards WFH; and (3) expectations that WFH would continue post-COVID-19. All of the conducted regression tests controlled for the age demographic variable, which was the only demographic characteristic other than company size that was shown to slightly correlate with the other variables. In these regression models, the variables are entered in the order they are listed above, which mirrors the forward regression method with the strongest predictors entered first. When the stepwise regression method was applied with the entire set of variables, only three out of seven were retained in the final model predicting WFH continuation after COVID-19 (WFH take-up during COVID-19, general attitude, and organizational WFH policies). As a set, the reliability and validity values of these retained variables met the customary cutoffs (composite reliability [CR] = 0.753, *p* = 0.385, df = 1; comparative fit index [CFI] = 1.000; root mean square error of approximation [RMSEA] = 0.000).

## 3. Results

### 3.1. Increase in Work-from-Home (WFH) Take-Up during COVID-19

The first research question of the present study was to verify if WFH has become more prevalent in South Korea during the government’s social distancing mandate to reduce the number of workers in the office. Comparisons of WFH take-up rates show that before COVID-19 only 13% of South Korean managers in the sample worked from home at least one day per week on average. Since the enactment of social distancing mandates, the managers in the sample reported that WFH take-up increased three-fold to 59% (Table 1). The mean number of days per week respondents personally worked from home increased from 0.34 prior to COVID-19 to 1.6 during COVID-19. Comparisons of background characteristics found that only two variables included in the analysis were statistically significantly correlated (*p* < 0.05) with current WFH take-up during COVID-19. These variables included Age and Size of Organization; however, the magnitude of the effect was relatively small (Eta^2^ = 0.042 and Eta^2^ = 0.091, respectively).

The results show a pattern of increased WFH take-up in organizations with more encouraging WFH policies. Among South Korean managers in the sample, 83% who WFH less than one day per week on average reported their organizations would only allow WFH during COVID-19 compared to 30% of those who WFH more than one day per week (Table 2). The reverse relationship was also demonstrated. Among respondents who WFH less than one day per week on average, 17% reported their organizations would continue to allow WFH after COVID-19 compared to 61% of those who WFH more than one day per week. Accordingly, reported organizational WFH policy was found to be statistically significantly correlated with WFH take-up (*p* < 0.001, Eta^2^ = 0.212) and it is expected continuation after COVID-19 subsides *(p* < 0.001, Eta^2^ = 0.265).

The results reveal an association between perceptions of job feasibility and WFH take-up, which confirms prior research studies [7,27]. Respondents who report higher WFH take-up report fewer job feasibility concerns on average (*p* < 0.001, Eta^2^ = 0.159). Over 62% of respondents who reported not working from home indicated that most employees’ jobs at their organizations could be done or could only partly be done at home (Table 3). Furthermore, the most common reason South Korean managers gave for not working from home was that their jobs required in-person contact (43%).

Eight out of nine items included in the management culture index were statistically significantly correlated with WFH take-up; however, these measures considered separately were not a major contributor to WFH outcomes for study’s participants (Table 4). When combined into an index, the effect size of the set of statistically significant variables within this index indicated moderate differences between the WFH subgroups (*p* < 0.001, Eta^2^ = 0.115). Therefore, this demonstrates that South Korean managers in the sample who reported more elevated autocratic workplaces were less likely to WFH. This result confirms recent previous research [41] that showed statistically significant yet weak effects for authoritarian leadership on support for WFH in the workplace.

### 3.2. Reconstructing General Attitude towards WFH

The second research question of this study was to explore the relationship between increased WFH take-up and reinforced positive attitudes towards WFH among South Korean managers in the sample. As predicted, results reveal rates of support for WFH were highest among those who WFH the most. Among respondents who WFH more than one day per week on average, 95% were in favor of it compared to 73% who WFH less than one day per week; a difference of 22%. This pattern was consistent across all of the factors measuring the general attitude construct (Table 5). For example, the results found WFH take-up had statistically significant yet moderate associations with how managers assessed the success of it for their organizations (*p* < 0.001, Eta^2^ = 0.167).

Associations were found between job satisfaction and the South Korean managers’ general attitudes towards WFH. Higher job satisfaction scores indicated a greater likelihood of favoring WFH (*p* < 0.001, Eta^2^ = 0.119), a result that once again confirms previous research [40]. Among survey respondents, 48% who were very satisfied with their jobs also strongly favored WFH compared to 16% of somewhat satisfied managers. A simpler pattern was demonstrated across all of the previously discussed variables associated with general attitude (Table 6). This result suggests that managers’ job satisfaction is likely to slightly bias their assessments of WFH outcomes.

Table 7 shows the results of three hierarchical linear multiple regressions models (one for each major dependent outcome variable) after controlling for age. Organizational WFH policies is the most statistically significant predictor of WFH take-up during COVID-19 (*p* < 0.001, R^2^ = 0.138). Thereupon, the WHF take-up variable uniquely contributed the most to explaining both general attitudes (*p* < 0.01, R^2^ = 0.070) and expected WFH continuation after COVID-19 (*p* < 0.001, R^2^ = 0.332) compared to other independent variables. The R^2^ scores for all the remaining independent variables (<0.100) indicated they were only slightly related to the aforementioned dependent variables and explained very little additional variance in the criteria for all three regression models or none at all if not statistically significant. The effects may be interpreted as follows: encouraging organizational WFH policies, job feasibility, high WFH take-up, and favorable attitudes towards WFH were found to have statistically significant unique positive effects on expected WFH continuation after COVID-19 subsides.

### 3.3. Expected WFH Continuation after COVID-19 Subsides

The third and final research question of this study focuses on whether managers’ forced WFH experiences due to social distancing measures would be maintained after COVID-19 subsides. Table 8 verifies earlier results that demonstrate WFH take-up contributes more than any other variable measured in this study to the prediction of anticipated WFH continuation in the future. Among South Korean managers in the sample, 77% who worked from home more than one day per week on average during COVID-19 reported that it was more likely to continue for all employees due to their organizations’ experience compared to 39% of those who did not WFH (*p* < 0.001, Eta^2^ = 0.292). Further analysis reveals that organizational WFH policy uniquely contributes more to managers’ forecasts WFH take-up for all employees (*p* < 0.001, R^2^ = 0.288) compared to their expectations of future WFH continuation for themselves (*p* < 0.001, R^2^ = 0.059) while controlling for all of the independent variables included in the earlier discussed regression models. A relatively stronger connection between organizational WFH policy and general WFH take-up among all employees during COVID-19 was also indicated (*p* < 0.001, R^2^ = 0.186). These results suggest that managers perceived organization WFH policy as having less influence on their own personal WFH decisions compared to other employees.

### 3.4. Graphic Model of Results

As shown in Figure 2 below, the results of the hierarchical linear multiple regression models were added to the proposed conceptual model to provide a concluding summary of the statistical analysis. Government-imposed WFH take-up had a unique positive effect on managers’ general attitude towards WFH and both variables combined together were more important than organizational policy for predicting future WFH continuation. The total effects of the aforementioned variables on expected WFH post-COVID-19 are highly statistically significant (*p* < 0.001, R^2^ = 0.443). These results answer the three research questions by demonstrating that increased WFH take-up during COVID-19 positively changed managers’ attitudes and would likely result in higher rates of WFH take-up among South Korean employees for the long term, but at lower levels compared to during the epidemic. As reported earlier (Figure 2), 44% of respondents expect to continue WFH after COVID-19, which is a tri-fold increase from the overall reported WFH take-up rate prior to COVID-19 of 13%; and managers anticipate continuing to WFH an average of 1.1 days per week after COVID-19 subsides.

## 4. Discussion of the Results

The first aim of this study was to determine if managers’ attitudes had been reconstructed as a result of their increased participation in WFH over the past two years and if the changes had an impact on their expectations to continue working from home in the future. As expected, this study confirms that WFH take-up in South Korea increased dramatically due to social distancing restrictions imposed during COVID-19 and this had unique statistically significant positive effects on both managers’ general attitudes towards WFH and their expectations they would be able to continue working from home post-COVID-19 (*p* < 0.01, R^2^ = 0.070 and *p* < 0.001, R^2^ = 0.330, respectively).

However, multiple hierarchical linear regression models showed that managers’ general attitude and organizational WFH policies were both relatively weaker predictors of their expectations (*p* < 0.05, R^2^ = 0.021 and *p* < 0.001, R^2^ = 0.059, respectively) compared to WFH take-up (*p* < 0.001, R^2^ = 0.330). Possibly, working from home an average of 1.6 days per week during COVID-19 falls short of representing a truly substantial shift in the working lives of managers, and thus their general attitudes towards WFH were only marginally impacted. Most South Korean employees are still going to the office for the most part and may only be required to WFH the minimum amount of time which is required to comply with government mandates. In this sense, the results of the present study are mixed; COVID-19 did and did not change the way South Korean managers work.

The Hierarchy of Effects (HOE) theory supposed that through their increased first-hand engagement with WFH managers will ‘think’ positively about it and therefore decide to ‘do’ it. An additional explanation for the weak relationship between attitudes and WFH take-up found in the present study may be that these two variables are not actually linearly related, as was assumed. In the current COVID-19 environment, employees are not ‘choosing’ WFH because they favor it over going to the office. Instead, organizations are requiring WFH out of an obligation to follow government mandates despite widespread and deep-seeded apprehensions about it. Managers are most likely behaving pragmatically; they are aware that their personal preferences for WFH have little effect on shaping organizational WHF policies. Rather, government mandates, organizational aims, and the expectations of top executives are probably understood to be more important considerations. In other words, South Korean managers in the sample understand that they do not decide whether they personally or their subordinates will WFH in the future, rather, it is top-level executives who make these decisions.

The present study accounted for many associated factors such as organizational culture, job satisfaction, and positive and negative trade-offs but there are likely many other unmeasured extraneous variables that might affect South Korean managers’ options, interest in and ability to WFH as well. These may include other social, cultural, and personal factors like individual circumstances that have more influence over the cognitive thinking and behaviors of managers. For example, the impact of WFH on the life domains of managers was not explored in this study. Other unconsidered factors are possible topics for further research that may require more in-depth research approaches.

Finally, the marginal relationship between increasing positive attitude towards WFH and the expectation of continued WFH practices in the future might be even lower in the present study because of South Korea’s distinctly orthodox corporate culture which reinforces the mainstream preference for full-time office work. However, the natural experiment imposed by COVID-19 showed that in the relatively short period of two years the managers experienced the benefits that WFH brings such as work–life balance, flexibility, and shorter commutes. Nonetheless, it appears that two years was not enough or the shift was not sizable enough to change long-held personal, organizational, and cultural structures that emphasize being present in the office and face-to-face interactions being very valuable in the work environment in South Korea.

## 5. Conclusions

The present study confirms that like countless employees around the world, most South Korean managers have worked at home at least intermittently during the COVID-19 pandemic, and for them, the shift to WFH has been very consequential. Despite deep-seated resistance to WFH in South Korea, approximately 60% of managers in the sample worked from home during COVID-19 at least one day per week on average, and nearly a half expect to continue doing so even after the crisis is over. The significant outcome of this study was to demonstrate that this considerable increase in experience with WFH has positively impacted, but not entirely reconstructed, their general attitudes towards WFH. This change is likely to reverberate among South Korean organizations and cause them to begin rethinking the employee experience, especially considering that social distancing mandates may stay in place for a long time to come.

It has been stipulated that after COVID-19, work would never be the same again, and the vast majority of managers in the present study who currently WFH (95%) do not want to go back to the office full time. Even though most of them expect WFH to end and the office-centric way of working will return, there is still a significant portion (39%) who believe their organizations will continue to allow WFH even after COVID-19 subsides. These results indicate that COVID-19 has created new expectations about the benefits of WFH for many South Korean organizations. Most likely, what is slowly emerging is a hybrid model for employees who are able to work remotely where the gains of part-time WFH are balanced with the need for social engagement through face-to-face interactions in the office.

Organizations become more aware of the interest in WFH among employees and as the approval of the hybrid model grows, corporate leaders in South Korea and around the world will need to deliberately manage the WFH experience to sustain confidence and better support employees. The present study has shown that managers report a range of positive and negative sentiments towards WFH. This result suggests a nuanced picture of the WFH experience. Different approaches to management will be needed to fit the diverse challenges and needs of individual employees and enable them to adjust to a new virtual workplace [45,46].

The present study shows that organizational WFH policy and management culture significantly influence WFH behaviors. As organizations recalibrate to adopt the hybrid model and adjust to the challenges of distributed work, they will establish new policies that will determine which jobs can be done remotely and govern how employees who WFH will be managed. Given the situation, the practical contribution of the present study is to help organizations and governments to understand the implications of their policy changes on the productivity, job satisfaction, motivations and social cohesion of their employees.

### Limitations and Future Work

The current study has several limitations—some of them are theoretical and some are data related. From the theoretical perspective, we assumed that managers would be rational and that changes in attitude would produce changes in behavior. There are known limitations of this hierarchical theory, in which experience (the environment) conditions attitudes and attitudes produce desired behaviors. These assumptions are rooted in behaviorist psychology and may not be realistic, but are often used in studies that are based on quantitative research methods. From a practical and data perspective, conducting a point-in-time survey with a limited sample size might not be completely representative of South Korean management attitudes. There is also the risk that error might have been introduced into the results of the present study due to respondents’ confirmation bias for certain questions like the managerial culture that were written from a critical perspective. 

Future research should focus on identifying other factors that might influence attitudes such as political inclinations and family situations, relating it to the literature on the use of digital technologies for personnel management [47,48,49]. In addition, interviewing upper managers who determine company policy would provide insight into the WFH trade-offs perceived by the higher-level executives who actually determine organizational policy. Furthermore, our data capture only expectations and future follow-up research performing longitudinal analysis could verify if those expectations are realized.

## Figures and Tables

**Figure 1 behavsci-11-00163-f001:**
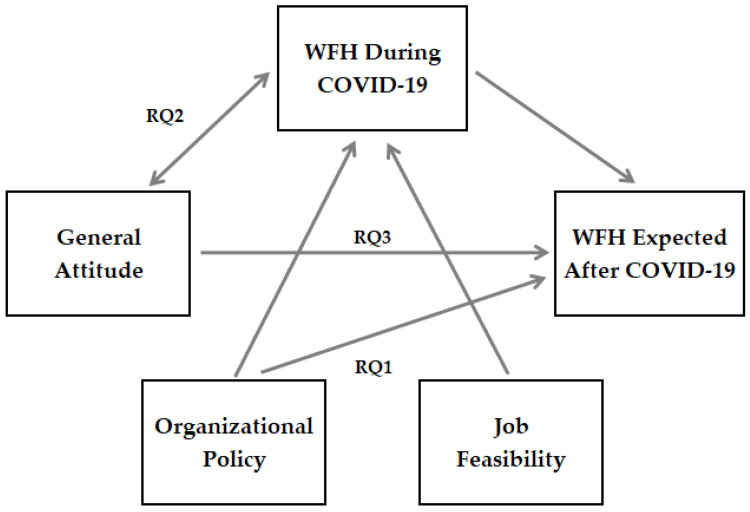
Proposed Conceptual Model.

**Figure 2 behavsci-11-00163-f002:**
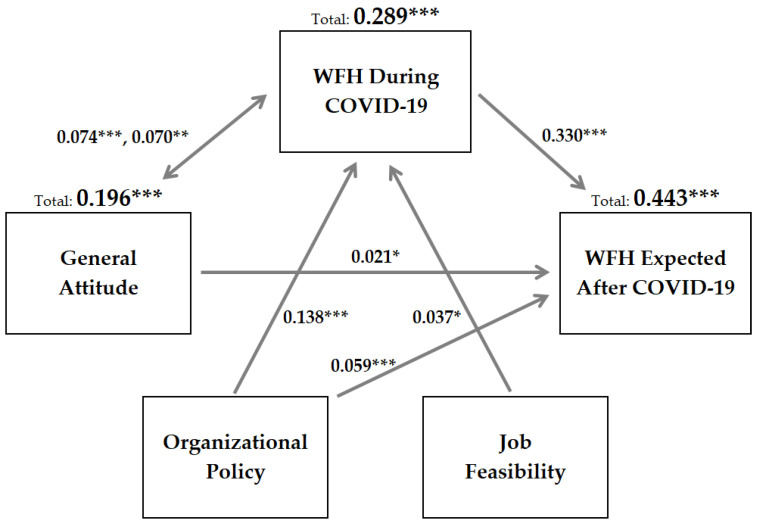
Final Conceptual Model. Note: Multiple hierarchical regression R-square Change and Total Adjusted R-square (* *p* < 0.05; ** *p* < 0.01; *** *p* < 0.001).

**Table 1 behavsci-11-00163-t001:** Frequencies of WFH Take-Up Before, During and Expected After COVID-19 by Demographic Subgroups.

		One or More Days WFH Per Week		
Time	*n*	Before COVID-19 (%/Mean)	During COVID-19 (%/Mean)	Expected after COVID-19 (%/Mean)
Total	229	13%/0.3	59%/1.6	44%/1.1
Gender				
Male	134	11%/0.2	56%/1.5	47%/1.2
Female	91	16%/0.5	62%/1.7	39%/1.0
Age				
18–29 Years Old	29	7%/0.3	59%/1.8 *	45%/1.3 **
30–39 Years Old	42	14%/0.4	67%/2.0 *	54%/1.6 **
40–49 Years Old	57	16%/0.4	70%/1.8 *	53%/1.2 **
50+ Years Old	97	13%/0.3	48%/1.2 *	33%/0.7 **
Married				
Yes	163	13%/0.3 15%/0.5	59%/1.6 55%/1.7	42%/1.0 47%/1.3
No	60			
Education Level				
High School Graduate	13	15%/0.5	38%/1.1	30%/0.8
2-year College	9	22%/0.4	44%/0.9	33%/0.7
4-year University	126	12%/0.4	65%/1.7	47%/1.2
Masters degree	61	13%/0.3	51%/1.4	46%/1.0
PhD or Doctoral Degree	16	19%/0.3	60%/1.6	27%/1.0
Length of Employment				
Less than 6 months	16	25%/0.9	63%/2.2	50%/1.9 ***
6–11 months	19	16%/0.6	58%/2.0	53%/2.1 ***
1–4 years	61	10%/0.3	52%/1.6	38%/0.9 ***
4–8 years	20	15%/0.2	55%/1.4	26%/0.6 ***
More Than 8 years	107	13%/0.3	61%/1.4	47%/1.0 ***
Employees Supervised				
One, Just Myself	83	17%/0.4	64%/1.8	50%/1.2
Under 5 People	73	10%/0.3	56%/1.5	42%/1.1
5–19 People	41	10%/0.3	56%/1.4	33%/0.8
20–49 People	17	12%/0.2	40%/1.0	33%/0.6
50 or More	10	20%/0.5	67%/1.7	63%/1.2
Size of Organization				
One, Just me	2	50%/1.0	50%/1.8 **	50%/1.5 *
Fewer than 10	40	18%/0.4	44%/1.2 **	29%/0.9 *
10–49 People	37	11%/0.2	25%/0.9 **	31%/0.6 *
50–99 People	28	25%/0.9	59%/2.2 **	46%/1.5 *
100–499 People	34	15%/0.4	76%/2.0 **	61%/1.3 *
500–1000 People	20	10%/0.4	65%/2.2 **	60%/1.8 *
More Than 1000 People	62	6%/0.1	76%/1.6 **	45%/0.9 *

Note: Correlations (* *p* < 0.05; ** *p* < 0.01; *** *p* < 0.001).

**Table 2 behavsci-11-00163-t002:** Frequencies of Organizations’ WFH Policies by WFH Take-Up Subgroups.

Organizational Policy	*n*	1	2	3	4
Time WFH DuringCOVID-19	194	41%	20%	25%	14%
0 (None)	36	64%	19%	11%	6%
1–3 Days Per Month	40	45%	23%	23%	10%
1 Day Per Week	44	59%	14%	20%	7%
2 Days Per Week	20	35%	20%	35%	10%
3 Days Per Week	22	14%	23%	36%	27%
4 Days Per Week	15	7%	27%	40%	27%
5 Days Per Week	17	6%	24%	29%	41%
Eta Squared	0.212				
*F*-value	8.40				
Sig.	<0.001				

Note: 1. Employees can WFH only when it is recommended by the government; 2. Employees will WFH until COVID-19 subsides even when it is NOT recommended by the government and afterwards they will return to the office; 3. After COVID-19 subsides employees will be allowed to continue WFH, but not as much as now during COVID-19; 4. After COVID-19 subsides, employees will be allowed to continue WFH the same levels as now during COVID-19.

**Table 3 behavsci-11-00163-t003:** Frequencies of Job Feasibility by WFH Take-Up Subgroups.

Job Feasibility	*n*	1	2	3	4
Time WFH During COVID-19	217	16%	32%	40%	13%
0 (None)	44	36%	36%	23%	5%
1–3 Days Per Month	45	16%	36%	36%	13%
1 Day Per Week	47	13%	32%	49%	6%
2 Days Per Week	21	5%	57%	38%	–
3 Days Per Week	24	8%	21%	46%	25%
4 Days Per Week	17	–	12%	65%	24%
5 Days Per Week	19	11%	16%	37%	37%
Eta Squared	0.155				
*F*-value	13.04				
Sig.	<0.001				

Note: 1. Most employees’ jobs cannot be done remotely; 2. Most employees can only do part of their jobs from home; 3. Most employees can work from home, but with difficulty; 4. Most employees can work from home without difficulty.

**Table 4 behavsci-11-00163-t004:** Means Comparison Analysis of Association between Management Culture and WFH Take-Up Subgroups.

Management Culture	*n*	Index Score	1	2	3	4	5	6	7	8
Time WFH During COVID-19	221	2.6	2.8	3.0	2.7	3.0	3.2	2.9	3.0	2.4
0 (None)	47	2.7	2.7	2.5	2.7	2.6	2.8	2.4	2.7	2.2
1–3 Days Per Month	45	2.9	2.7	2.9	2.4	2.7	2.9	2.9	2.8	2.2
1 Day Per Week	47	2.9	2.6	3.0	2.6	3.3	3.3	2.9	2.9	2.4
2 Days Per Week	22	2.8	2.7	3.3	2.6	2.9	3.3	3.1	3.2	2.3
3 Days Per Week	24	3.4	2.8	2.8	2.6	2.8	3.2	3.0	2.6	2.2
4 Days Per Week	17	3.3	3.5	3.5	3.2	3.4	3.9	3.3	3.4	3.1
5 Days Per Week	19	2.8	3.4	3.4	3.3	3.4	3.6	3.3	3.7	2.6
Eta Squared		0.115	0.095	0.094	0.092	0.091	0.086	0.077	0.069	0.068
*F*-value		4.66	3.74	3.63	3.62	3.57	3.32	2.95	2.61	2.57
Sig.		<0.001	0.001	0.002	0.002	0.002	0.004	0.009	0.019	0.020

Note: 1. Employees try hard to keep a distance from their managers; 2. Managers always have the last say in meetings; 3. Employees feel tense when they are with their managers; 4. Employees’ job contents are assigned directly by their managers; 5. Employees have to rely directly on their managers for assistance to complete work; 6. Managers make all decisions in their teams, whether they are important or not; 7. In the minds of managers, they believe an ideal subordinate is one who always obeys their wishes; 8. Managers ask employees to obey their instructions completely.

**Table 5 behavsci-11-00163-t005:** Means Comparison Analysis of Association between General Attitude towards WFH and WFH Take-Up Subgroups.

General Attitude	*n*	1	2	3	4	5	6
Time WFH During COVID-19	221	2.5	2.6	2.5	2.0	4.4	2.2
0 (None)	46	3.0	2.4	2.3	2.3	4.9	2.4
1–3 Days Per Month	45	2.8	2.4	2.4	2.2	4.5	2.5
1 Day Per Week	47	2.8	2.7	2.5	2.1	4.2	2.2
2 Days Per Week	22	2.5	2.3	2.5	1.9	4.5	2.1
3 Days Per Week	25	2.3	2.9	2.9	1.7	3.9	1.9
4 Days Per Week	17	2.1	3.1	2.8	1.8	4.0	1.8
5 Days Per Week	19	1.7	3.2	2.8	1.5	4.0	1.9
Eta Squared		0.167	0.124	0.109	0.107	0.097	0.062
*F*-value		6.25	4.92	4.38	4.25	3.80	2.20
Sig.		<0.001	<0.001	<0.001	<0.001	0.001	0.045

Note: 1. Success of WFH for Organization; 2. Productivity while WFH; 3. Trust in Employees to WFH; 4. In Favor of WFH; 5. Positive and Negative Tradeoffs of WFH; 6. Colleagues’ Reaction to WFH.

**Table 6 behavsci-11-00163-t006:** Means Comparison Analysis of Association between Job Satisfaction and General Attitude towards WFH.

Job Satisfaction	*n*	1	2	3	4	5	6
Time WFH During COVID-19	224	2.5	2.7	4.4	2.5	2.2	2.0
Very Satisfied	41	1.8	3.1	3.7	2.9	1.7	1.7
Somewhat Satisfied	147	2.6	2.7	4.4	2.5	2.2	2.0
Somewhat Dissatisfied	33	3.1	2.1	5.1	2.2	2.8	2.4
Very Dissatisfied	3	5.0	1.0	5.9	1.3	4.0	3.3
Eta Squared		0.202	0.195	0.168	0.154	0.135	0.119
*F*-values		16.46	17.51	14.71	13.32	10.81	9.82
Sig.		<0.001	<0.001	<0.001	<0.001	<0.001	<0.001

Note: 1. Success of WFH for Organization; 2. Productivity while WFH; 3. Positive and Negative Tradeoffs of WFH; 4. Trust in Employees to WFH; 5. Colleagues’ Reaction to WFH; 6. In Favor of WFH.

**Table 7 behavsci-11-00163-t007:** Hierarchical Linear Multiple Regression Models.

Factor	Time WFH during COVID-19	In Favor of WFH	Time WFH Expected after COVID-19
Control: Age	0.023 *	0.069 ***	0.043 **
Time WFH During COVID-19	---	0.077 ***	0.371 ***
In Favor of WFH	0.033 *	---	0.100 ***
Organizational WFH Policy	0.192 ***	0.029 ***	0.233 ***
Time WFH Before COVID-19	0.139 ***	0.026 ***	0.140 ***
Management Culture	0.124 **	0.093 ***	0.078 **
Job Satisfaction	0.090 **	0.104 ***	0.039 **
Job Feasibility	0.060 **	0.071 ***	0.118 ***

Note: Simple linear regression statistics calculated for the separate effects of each independent variable on each dependent variable while controlling for age (* *p* < 0.05; ** *p* < 0.01; *** *p* < 0.001). Adjust R-square is calculated for the total effects of all independent variables as a set on the major dependent outcome variable in each hierarchical regression model.

**Table 8 behavsci-11-00163-t008:** Frequencies of Likelihood of WFH Continuation Post-COVID-19 by WFH Take-Up Subgroups.

Likelihood Organization will Continue WFH Post-COVID-19	*n*	Much More Likely Now	Somewhat More Likely Now	No Change	Somewhat Less Likely Now	Much Less Likely Now
Time WFH During COVID-19	196	17%	33%	14%	11%	25%
0 (None)	25	4%	36%	36%	12%	12%
1–3 Days Per Month	42	12%	29%	17%	14%	29%
1 Day Per Week	46	2%	13%	13%	22%	50%
2 Days Per Week	22	18%	41%	9%	9%	23%
3 Days Per Week	25	24%	52%	12%	0%	12%
4 Days Per Week	17	47%	47%	0%	6%	0%
5 Days Per Week	19	47%	37%	5%	0%	11%
Eta Squared	0.292					
*F*-value	13.00					
Sig.	<0.001					

## Data Availability

The dataset for the present study is available on request from the authors.

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
