# Peer review of "Reconstructing Attitudes towards Work from Home during COVID-19: A Survey of South Korean Managers"

_behavsci, 2021, doi:10.3390/bs11120163_

Round 1

Reviewer 1 Report

This study analyzed the change of managers’ attitude towards WFH during the COVID-19 and well-constructed to address the issues. However, I believe some parts of this paper should be clarified to be accepted for publication.
The authors surveyed ‘managers’, yet did not identified “who are the managers?”. Depends on the organizational size, there might be different levels of managerial positions. As authors presented in the Table 1, there are 83 managers supervising one employee or ‘just myself’, and there is also one-person organization (2), and managers who are 18-29 years old (29). 

In the survey questionnaire, there is an item, 5 days per week, in ‘time WFH during COVID-19’. Is this 5 working-days per week? If so it is hard to believe a manager works at home even during the COVID-19 period when we consider Korea’s a hard-working environment, as the author described.   

On page 6, The authors mentioned the control variables are gender, age, marital status, educational attainment, and annual household income. Yet the authors controlled only age variable (page 13). The authors need to explain the reason.

Reviewer 2 Report

Comments and Suggestions for Authors

Dear Authors, thank You for so interesting research.

  1. General concept comments.

The article is written on the relevant topic and is well structured as well as logically proved.

However, the structure of the article is quite unusual when the developed approach is described in the Introduction. Nevertheless, the second section discloses the theory of the methods used.

Please kindly make better for understanding the way how all three research questions are studied. Could You please in the end of the resulting section clearly explain in some paragraphs how the researchers prove the testability of three propositions on the basis of the developed model. The explanation is necessary because it allows to avoid methodological inaccuracies.

I’d recommend the authors should move the most part of the discussion section into the end of the result section. Then authors could add the sixth section with the limitations of the model to the discussion section for improving the structure of the article.

  1. Scientific Novelty.

Please kindly describe the scientific contribution to the theory in conclusion. For example, the researchers suggest the approach allowing to extend the Hierarchy of Effects theory on the basis of additional factors’ impact on individual’s behavior during the work-from-home (culture, job satisfaction, positive and negative tradeoffs) newly explained in the article.

  1. General questions.

I think the researchers should carefully describe in the Discussion section the connection between the topic of the Article and the issues for future research. The authors could offer to build the logical bridge to the digital personnel management concept after considering "other factors that might influence attitudes such as political inclinations and family situations".

For this purpose, it could be recommended to add to the literature review some references regarding the wider understanding the use of digital technologies for personnel management:

Barykin, S.Y., Kapustina, I.V., Valebnikova, O.A., Valebnikova, N.V., Kalinina, O.V., Sergeev, S.M., Camastral, M., Putikhin, Y., Volkova, L. Digital Technologies For Personnel Management: Implications For Open Innovations (2021) Academy of Strategic Management Journal, 20 (SpecialIssue2), pp. 1-14. https://www.scopus.com/inward/record.uri?eid=2-s2.0-85107756718&partnerID=40&md5=c3b5d3f42a19286b373fbc28d21fd73b

  1. Ethics statements and data availability statements.

In view of practical significance of the developed approach with the data used, the authors have shown the correct understanding of both the recommended guidelines of the Committee on Publication Ethics (https://publicationethics.org/) and Institutional Review Board Written Procedures: Guidance for Institutions and IRBs (2018).
